# Control of carbon monoxide dehydrogenase orientation by site-specific immobilization enables direct electrical contact between enzyme cofactor and solid surface

Stacy Simai Reginald[1], Hyeryeong Lee[1], Nabilah Fazil[1], Basit Sharif[1], Mungyu Lee[1], Min Ji Kim[1], Haluk Beyenal [2] & In Seop Chang [1✉]

Controlling the orientation of redox enzymes on electrode surfaces is essential in the development of direct electron transfer (DET)-based bioelectrocatalytic systems. The electron transfer (ET) distance varies according to the enzyme orientation when immobilized on an electrode surface, which influences the interfacial ET rate. We report control of the orientation of carbon monoxide dehydrogenase (CODH) as a model enzyme through the fusion of gold-binding peptide (gbp) at either the N- or the C-terminus, and at both termini to strengthen the binding interactions between the fusion enzyme and the gold surface. Key factors influenced by the gbp fusion site are described. Collectively, our data show that control of the CODH orientation on an electrode surface is achieved through the presence of dual tethering sites, which maintains the enzyme cofactor within a DET-available distance (<14 Å), thereby promoting DET at the enzyme–electrode interface.

[1] School of Earth Sciences and Environmental Engineering, Gwangju Institute of Science and Technology, 123 Cheomdan-gwagiro, Buk-gu, Gwangju 61005, Republic of Korea. [2] The Gene and Voiland School of Chemical Engineering and Bioengineering, Washington State University, Pullman, WA 99164, United States of America. ✉email: ischang@gist.ac.kr

Efficient electron transfer (ET) between biological redox enzymes and electrode surfaces at the enzyme–electrode interface is crucial for the development of bioelectrocatalytic systems, including enzyme-based biosensors, enzymatic biofuel cells, and enzymatic electrosynthesis[1–4]. Two types of ET mechanism are known, namely direct electron transfer (DET), in which electrons are exchanged directly with the electrode surface, and mediated electron transfer (MET), in which small mobile redox mediators are used as electron shuttles between the enzyme and the electrode surface[5–8]. DET is often favored over MET because DET avoids intermediate ET steps by self-exchange reactions, eliminates the use of redox mediators that are often associated with toxicity[9] and thermodynamic losses[6], and increases the interfacial ET rate[1,10]. Among the various factors contributing to the efficiency of DET at the enzyme–electrode interface, it should be considered that the ET distance must be within ~14 Å[1,11–14], since beyond this distance, the rate of ET rapidly approaches zero, as explained by the Marcus theory[6,15]. In addition, the enzyme should retain its bio-functionality when immobilized on the electrode surface[16,17], and the formation of a stable electrode-immobilized enzyme is necessary[18]. Importantly, the enzyme on the electrode surface should adopt a so-called "electroactive" configuration, i.e., it should be within the distance that permits DET between the enzyme and the surface[11,19,20] and it should have an orientation that facilitates substrate accessibility[21].

In this context, the key challenge in the pursuit of efficient DET is the size and complexity of the three-dimensional (3D) structure of the protein that carries out direct electrical communication at the enzyme–electrode interface. All enzymes consist of a polypeptide backbone arranged in secondary and tertiary structures, and they contain redox cofactors that are either metal complexes or organic molecules bound to a specific site in a pocket or cleft on the enzyme surface[22]. In general, redox enzymes possess average hydrodynamic diameters ranging from 55 to 150 Å (40–850 kDa), with one or more redox centers[23] that are electrically inaccessible because their active sites are buried deep beneath the surface of the protecting protein shell[24]. In a natural system, the physiological distances between the edges of the redox centers are within ~14 Å, which permits efficient electron tunneling[12]. However, electrons must often be transferred over distances >14 Å, which is typically accomplished by multistep tunneling through chains of redox centers individually positioned within the 14 Å boundary[13]. In contrast, in a bioelectrocatalytic system, the electrode is the unnatural electron acceptor or donor;[25,26] thus, direct electrical wiring between enzymes and the electrodes that they are confined to is rare because of the large sizes of the enzymes, ultimately resulting in inaccessibility to the redox center[1,23]. To date, many studies have reported successful efforts to achieve DET between enzymes and electrodes, including (1) electrode modification using nanomaterials such as nanoparticles and nanotubes[27–30], since the sizes and shapes of these species facilitate enzyme adsorption, loading, and wiring[7,10]; (2) the use of self-assembled monolayers (SAMs) that serve as bridges between redox enzymes and solid electrode surfaces[31–35]; (3) protein engineering efforts that mimic naturally evolved ET proteins by fusing the cytochrome domain to the enzymes[25,36–39]; and (4) the employment of site-directed mutagenesis to achieve site-directed immobilization on the electrode surface[1,40–45]. However, these strategies may not allow either the distance between the enzyme cofactor and the electrode surface or the enzyme orientation to be controlled sensitively because of the inherent complexity of the 3D enzyme structure, ultimately resulting in an unsatisfactory interfacial ET efficiency. Previously, our group described a sophisticated immobilization technology that used the fusion of a gold-binding peptide (gbp) composed of

12 amino acids (LKAHLPPSRLPS)[46] as a molecular linker to flavin adenine dinucleotide (FAD)-dependent glucose dehydrogenase (GDH)[43]. This system exhibited a high binding affinity and a preserved enzyme bioactivity, in addition to allowing the formation of a uniform enzyme monolayer on the gold electrode, ultimately facilitating DET at the enzyme–electrode interface[43]. Although the DET rate was drastically enhanced when the gbp was fused to the C-terminus of the α-subunit of FAD-GDH[43], the orientation flexibility of the enzyme was considered when only a single tethering site was available[47,48], resulting in a non-specifically controlled ET distance. Subsequently, the possibility of introducing two or more gbp-fusion sites to control the distance between the enzyme cofactor and the immobilization site (i.e., the enzyme–electrode interface) sensitively was considered to allow control of the enzyme orientation and the DET rate.

For the present work, the effect of the distance between the immobilization site and the enzyme cofactor on the ET kinetics at the enzyme–electrode interface was investigated by immobilizing a model enzyme via single or dual tethering sites. For this purpose, an enzyme was designed and engineered to fuse the gbp at specific sites, to generate various immobilization sites, and control the distance between the immobilization site and the cofactor (i.e., the only variable parameter to consider). Because of the length of the gbp, selection of the fusion site needed to be carefully considered to avoid disrupting the catalytic pocket or the substrate-binding site. More specifically, a model enzyme, namely carbon monoxide dehydrogenase (CODH) from *Hydrogenophaga pseudoflava*, was employed, since it is a unique example of an enzyme that reacts with a small gaseous substrate (CO = 28.01 gmol$^{-1}$), besides having a resolved crystal structure (PDB ID:1FFV) that allows structure-guided selection for further protein engineering. In addition, the single L-subunit of this model enzyme has been confirmed to exhibit a CO oxidation activity even in the absence of the M- and S- subunits[49], thereby allowing a system to be constructed with fewer determining factors (i.e., only the gbp-fusion site(s) differed). A local meta-threading server (LOMETS)[50,51] was used to analyze the enzyme structure to select gbp-fusion sites that were freely exposed to the environment without any restriction on the structural conformation or any potential interference between the enzyme catalytic domain and substrate-binding pockets.

To determine the effect of the gbp-fusion site on the enzyme orientation and ET distance, we initially sought to estimate the distance between the enzyme cofactor and the immobilization site (the site at which the electrode and enzyme are in direct contact), to determine the enzyme assembly characteristics, and to verify the binding affinities toward the gold surface. Finally, using electrochemical techniques to evaluate the ET properties at the enzyme–electrode interface, we found dramatically different ET rates depending on the gbp-fusion site. Ultimately, we demonstrate a promising platform technology for controlling the enzyme orientation with site-specific immobilization to shorten the ET distance across the enzyme–electrode interface and expect that it will be of particular importance in the fabrication of DET-based bioelectrocatalytic systems.

## Results

### Design of native and synthetic CODH-Ls with single and dual gbp-fusion sites for surface immobilization
Based on the LOMETS analysis of distance and the contact map for selecting the gbp-fusion site on CODH-L (see Fig. 1a, b, respectively), residue 1 (N-terminus) and residue 803 (C-terminus) were considered to be ideal for gbp-fusion because they have no contact with the catalytic domain and are solvent-exposed. Thus, native CODH-L was overexpressed, purified, confirmed by sodium

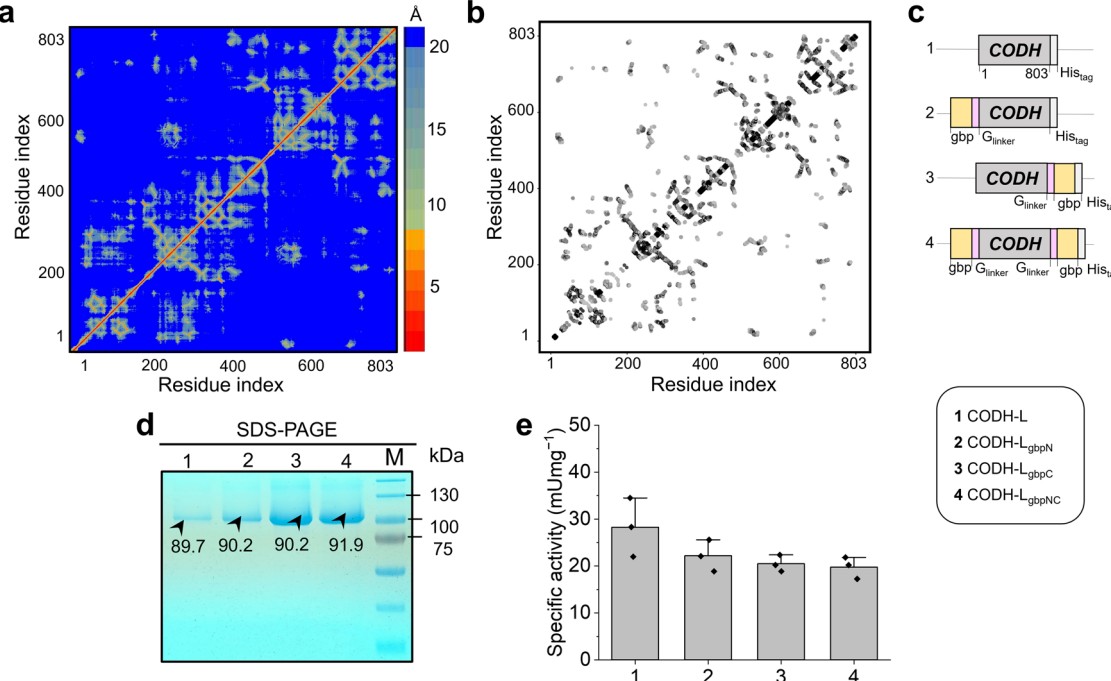

**Fig. 1 Enzyme structure analysis and results for the produced recombinant enzymes. a** The distance map and **b** contact map analysis for the native CODH-L structure derived from LOMETS. The color scale in the distance map represents a distance of 1 to >20 Å. For the contact map, each dot represents a residue pair with predicted contact. **c** The various plasmid constructs used in this study. **d** SDS-PAGE of the purified native and synthetic CODH-Ls. **e** Specific activity of the native and synthetic CODH-Ls. The bar graph showing mean and standard error mean (SEM), with data points ($n = 3$ independent replicates).

dodecyl-sulfate polyacrylamide gel electrophoresis (SDS-PAGE), and tested for its CO oxidation activities. Subsequently, the obtained recombinant CODH-L was used as a template for further genetic modification, shown in Fig. 1c. The SDS-PAGE results presented in Fig. 1d and Supplementary Fig. 1 show similar molecular masses for all variants at ~90 kDa, which suggests uniform glycosylation of the native and synthetic CODH-L. These results indicate that this technique does not modify the intrinsic enzyme activities, since the biocatalytic activities of the synthetic CODH-Ls were highly preserved (Fig. 1e), thereby rendering the gbp-fusion site(s) (i.e., the immobilization site(s)) the only variable to consider. The predicted structures of the synthetic CODH-Ls were then generated by iterative threading assembly refinement (I-TASSER)[52–54], and the structures with the highest confidence values (C-score) are shown in Fig. 2. Using various tools, the root mean square deviation (RMSD) of the synthetic enzymes against the native CODH-L (for all atoms) was calculated to be <2 Å, as tabulated in Supplementary Table 1, which indicates that no significant structural changes took place following the addition of gbp to the enzyme.

**Estimation of the distance between the enzyme active site and the electrode surface.** In a DET-based system, it is known that the enzyme bioactivity[16,17], the surface and solution chemistry, the substrate diffusion[12], and the ET distance at the enzyme–electrode interface[11–14] are among the factors that can affect the ET rate. In this study, CODH-L was designed and engineered to fuse the gbp at the N- or C- terminus, or at both termini, thereby generating various immobilization sites on the CODH-L. More specifically, the orientation of the enzyme molecules can be controlled depending on the immobilization site. As a result, the distance between the enzyme cofactor and the electrode surface, $d_{ET}$, is the only variable parameter to consider in this study, assuming that all other factors, such as the enzyme

bioactivity and the substrate diffusion, are negligible (or constant). It was therefore hypothesized that $d_{ET}$ is the main limiting factor of DET (considering that all other factors are constant); therefore, we initially estimated $d_{ET}$. More specifically, the estimated $d_{ET}$ based on the crystal structure of the native CODH-L is shown in Fig. 3 and Table 1. This estimation was carried out carefully considering the following points: 1) binding takes place at the terminus because the gbp is fused at the terminus, and 2) the size of the gbp is 1.3 kDa, which is only 1.49% of the size of CODH-L, and so becomes negligible in the estimation. Thus, to estimate $d_{ET}$ when the gbp is fused at the N- or C-terminus, the distance is taken under the condition that the longer enzyme axis is perpendicular (i.e., at 90°) to the electrode surface; therefore, the actual distance may be shorter, considering that the single tethering site used to bind the enzyme to the electrode surface may induce orientational flexibility (see Supplementary Movie).

Using PyMol, $d_{ET}$ was therefore estimated to be 29.56 ± 3.29 and 39.97 ± 1.16 Å for CODH-L$_{gbpN}$ and CODH-L$_{gbpC}$, respectively (Table 1). In the case, where the gbp was fused at both termini, it was hypothesized that the enzyme molecule would be positioned "lying down", i.e., with the longest axis parallel to the electrode surface. As the N- and C-termini of CODH-L are freely exposed, located away from the main domain, and extending in the same direction, a fusion of the gbp at both termini would potentially stabilize the enzyme further and limit the orientations that the enzyme could adopt, as indicated in Fig. 3. Based on these considerations, the distance to reach the cofactor would be ~10.27 ± 1.38 Å, which is <14 Å from the nearest residue facing the electrode, thereby rendering it a promising candidate for DET.

**Assembly properties of the native and synthetic CODH-Ls on the planar gold surface.** Atomic force microscopy (AFM) provides information regarding the morphological and topological

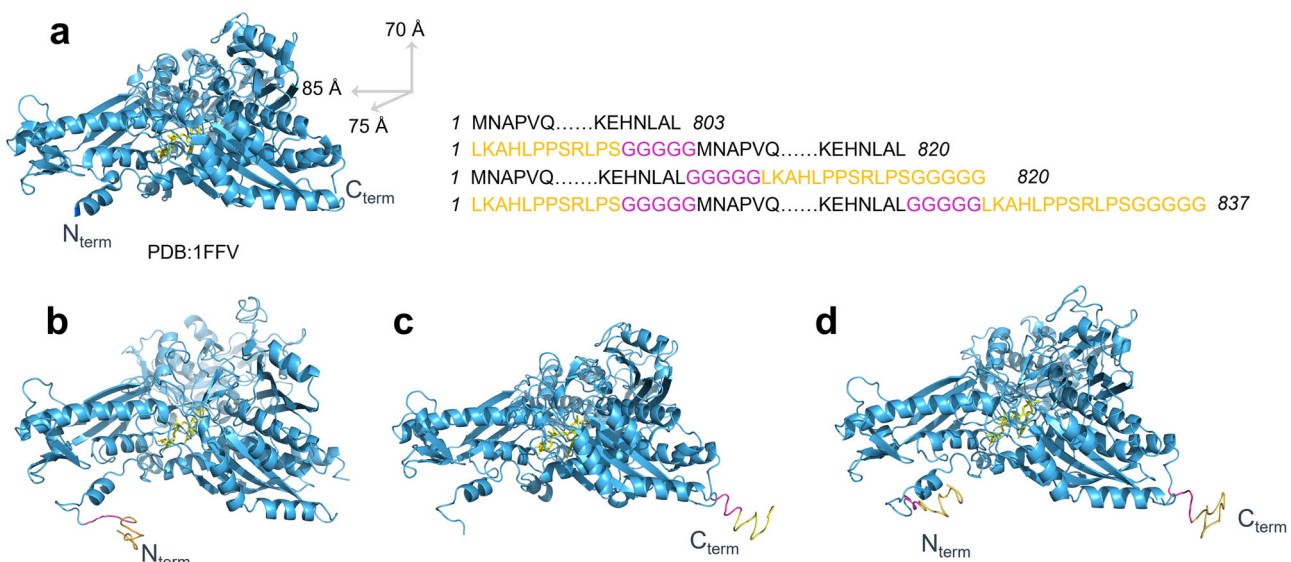

```
1  MNAPVQ......KEHNLAL  803
1  LKAHLPPSRLPSGGGGGMNAPVQ......KEHNLAL  820
1  MNAPVQ.......KEHNLALGGGGGLKAHLPPSRLPSGGGGG  820
1  LKAHLPPSRLPSGGGGGMNAPVQ......KEHNLALGGGGGLKAHLPPSRLPSGGGGG  837
```

**Fig. 2 Native CODH-L and the predicted structures of the synthetic enzymes.** The structures of **a** the native CODH-L, **b** CODH-L$_{gbpN}$, **c** CODH-L$_{gbpC}$, and **d** CODH-L$_{gbpNC}$, with the highest confidence value (C-score) predicted by I-TASSER, were aligned followed by structural superposition with the native CODH-L using PyMol.

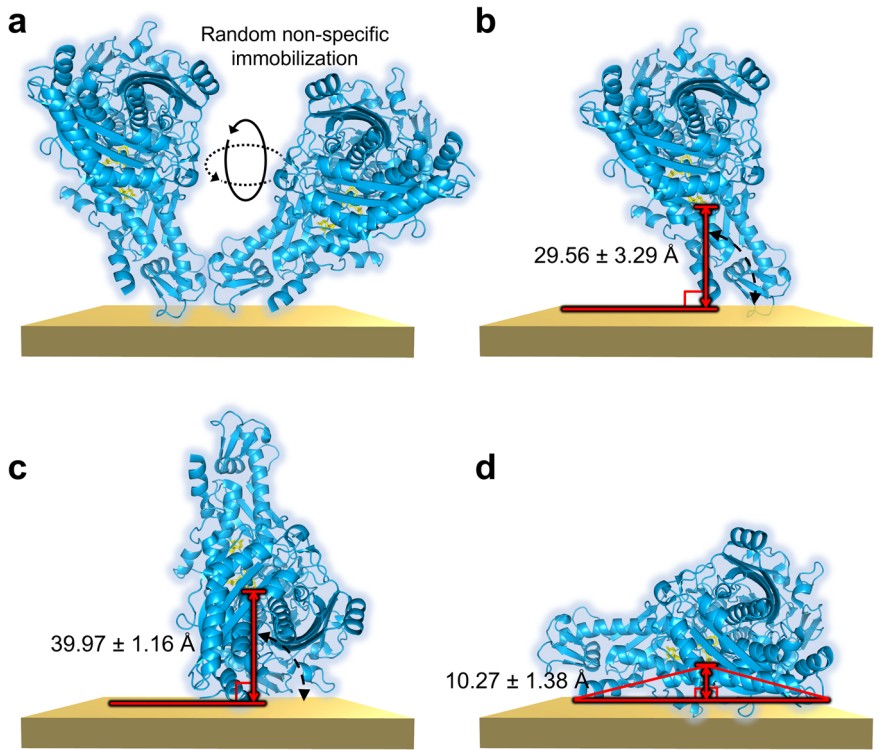

**Fig. 3 Estimation of the ET distance between the enzyme cofactor and the electrode surface.** A schematic illustration of the estimated distance between the Mo–Cu active site and the electrode surface, $d_{ET}$, is shown for **a** the native CODH-L, **b** CODH-L$_{gbpN}$, **c** CODH-L$_{gbpC}$, and **d** CODH-L$_{gbpNC}$.

features of the electrode surface, which in this case is modified with the enzymes. The bare gold substrate (treated only with buffer) measured ~2 nm in height and possessed −2 nm voids, indicating an atomically flat surface, thereby allowing the enzyme orientation on the gold surface to be examined (Supplementary Fig. 2). Considering that the CODH-L possesses dimensions of ~85 × 75 × 70 Å[55] and that the bare gold substrate contains nanometer-scale voids, it was assumed that a height (thickness) >5 nm indicates that the enzyme was immobilized on the gold

substrate (Fig. 4a–d, Supplementary Fig. 2a–e). This is because the shortest dimension of the enzyme is 70 Å (7 nm) and the voids are −2 nm; therefore, the measured height (thickness of enzyme on the gold surface) would be ~5 nm if the enzyme were immobilized such that its shortest axis was perpendicular to the gold surface. The cross-sectional height profile at a lateral distance of 5 μm of the selected line indicates that the native CODH-L was not well immobilized (Fig. 4a, lower panel). This was expected since the success of the immobilization process is highly

dependent on the formation of weak non-specific bonds between the enzyme and the gold surface.

It was found that the heights varied between 6 and 18 nm (±12 nm height difference), 6 and 9 nm (±3 nm), 6 and 12 nm (±6 nm), and 7 and 8 nm (±1 nm) for the native CODH-L, CODH-L$_{gbpN}$, CODH-L$_{gbpC}$, and CODH-L$_{gbpNC}$, respectively. These notable differences in the measured enzyme heights obtained by cross-sectional analysis imply differences in the orientations of the bound molecules, plausibly due to the binding site being only at one end in the cases of CODH-L$_{gbpN}$ and CODH-L$_{gbpC}$.

More specifically, the presence of the gbp at a single site renders the longest protein axis perpendicular to the gold substrate; i.e., it appears to be "standing" on the substrate surface. This results in relatively unstable enzyme molecules, which tend to tilt to find the lowest energy orientation, thereby leading to the observed differences in their heights. In contrast, the height was highly uniform in the case of CODH-L$_{gbpNC}$, indicating that the enzymes are immobilized in a near-uniform orientation when the gbp is fused to both termini. The corresponding height histogram, based on a more detailed analysis of the enzyme height on the gold substrate, is shown in Supplementary Fig. 3. The native and synthetic CODH-L, with the exception of CODH-L$_{gbpNC}$, demonstrated a wide distribution of heights, ranging from 6 to 18.9 nm, indicating the presence of more than one enzyme layer on the surface.

**Table 1 Estimated distances between the electrode surfaces and the Mo–Cu cofactor, $d_{ET}$.**

| Gbp-fusion site | Construct information | Cofactor-electrode distance, $d_{ET}$ (Å) |
|---|---|---|
| Residue 1 | CODH-L$_{gbpN}$ | 29.56 ± 3.29 |
| Residue 803 | CODH-L$_{gbpC}$ | 39.97 ± 1.16 |
| Residue 1 and 803 | CODH-L$_{gbpNC}$ | 10.27 ± 1.38 |

Data shown are mean ± s.d for three shortest ET distance (i.e., distance between cofactor and solid support).

Notably, the height distribution of CODH-L$_{gbpNC}$ ranged from 6 to 14 nm, and the histogram shows that >45% of the immobilized protein had a height of 6–6.9 nm, indicating a uniform adsorption geometry for the CODH-L$_{gbpNC}$ molecules. It was, therefore, speculated that the presence of two freely exposed immobilization sites on CODH-L$_{gbpNC}$ allowed the enzyme molecules to be positioned stably in an ideal orientation, i.e., "lying down" on the gold surface with the longer protein axis parallel to the gold surface, ultimately resulting in a uniform adsorption geometry. In addition, following modification, the surface roughness was found to have increased compared to that of the bare substrate, again reinforcing the hypothesis that the enzymes were successfully immobilized on the substrate (Supplementary Table 2), since the roughness is related to the protein concentration on the substrate surface.

**AFM analysis following the immobilization of native and synthetic CODH-L on a screen-printed gold electrode.** Supplementary Fig. 4 shows the AFM images and the 3D-surface profiles of the bare screen-printed gold electrode (SPGE), DRP-250 AT, and the enzyme-immobilized SPGE. As can be seen in the image of the bare SPGE, the surface is not perfectly flat, since significant voids and raised regions are present on the micrometer scale. For the native CODH-L immobilized on the electrode, there were no notable differences in the 3D image or the surface height compared to those of the bare SPGE. This was attributed to the reliance of the native CODH-L to take part in weak non-specific binding with the gold electrode surface, and so it can easily desorb from the electrode surface.

In contrast, significant differences were observed for the synthetic CODH-Ls bearing the gbp at the N- and/or C-terminus, with the surface now appearing to be flatter and the voids becoming less apparent. This can be attributed to the fact that the immobilized enzyme fills the deep voids, so that the peak-to-valley height decreases. This is especially prominent in the case of

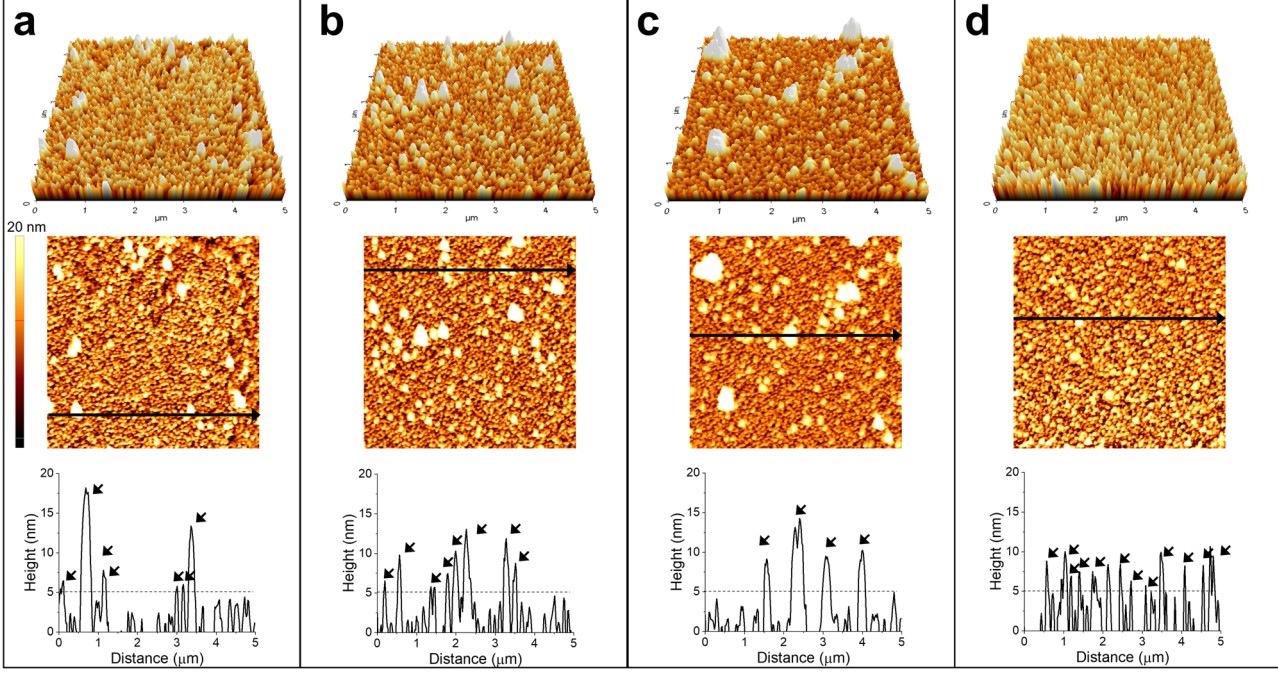

**Fig. 4 Assembly properties of the enzymes on the planar gold surface.** AFM images for **a** the native CODH-L, **b** CODH-L$_{gbpN}$, **c** CODH-L$_{gbpC}$, and **d** CODH-L$_{gbpNC}$ immobilized on the planar gold surface. The corresponding cross-sectional profiles along the indicated arrows are shown in the lower panels. The arrows indicate the location of the enzyme on the surface.

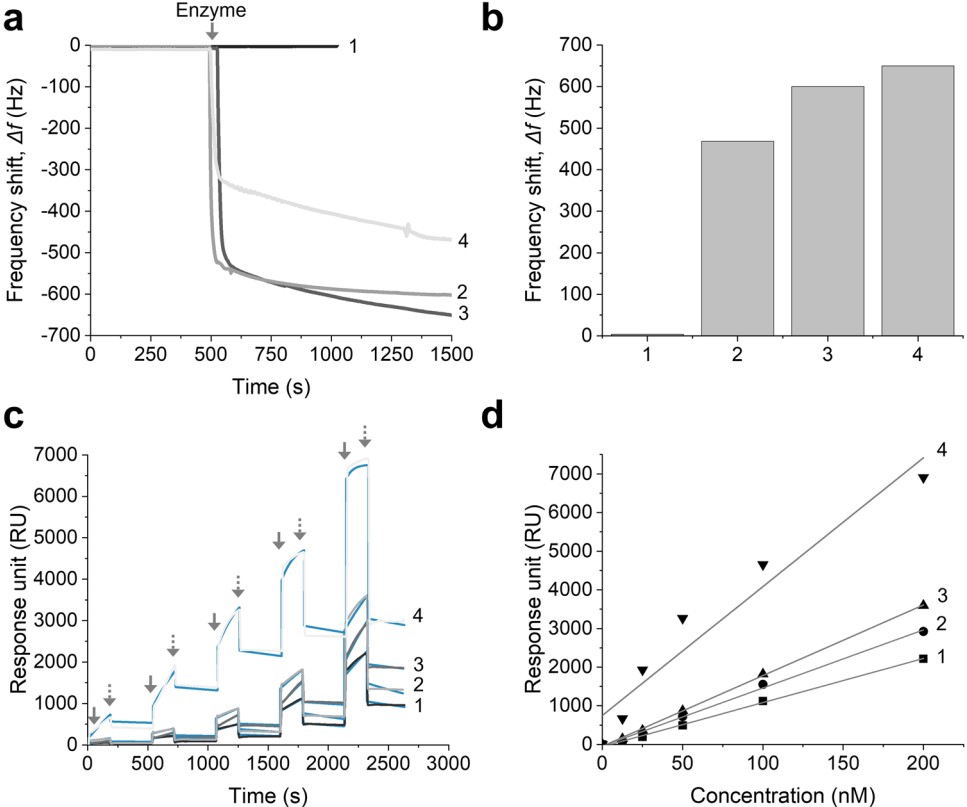

**Fig. 5 Gold-binding properties of the native and synthetic enzymes. a** Gold-binding activities of the native CODH-L and of the synthetic constructs as determined by QCM analysis (the arrow indicates the injection of 1 μM enzyme solution). **b** Comparison of the frequency shift in **a**. **c** Changes in the SPR refractive index as a function of time for enzyme adsorption on the gold surface at enzyme concentrations of 12.5, 25, 50, 100, and 200 nM (the solid arrows indicate enzyme injection while the dashed arrows indicate buffer injection). The blue traces represent the global fit of the data to a 1:1 binding model. **d** Titration curve obtained from plotting the binding response against the enzyme concentration. Key: 1, 2, 3, and 4 refer to CODH-L, CODH-L$_{gbpN}$, CODH-L$_{gbpC}$, and CODH-L$_{gbpNC}$, respectively.

CODH-L$_{gbpNC}$, in which the surface becomes flattened, indicating a high enzyme coverage on the SPGE surface.

Because of the presence of significant voids and raised regions on the micrometer scale, it was not possible to observe how the enzyme molecules were oriented on the SPGE. However, in the AFM study into the SPGE with a structured topography and an atomically flat gold surface, it was demonstrated that gbp-fused enzymes are highly specific for gold surfaces, regardless of their surface properties, as all three fusion enzymes were successfully immobilized on the gold surface.

**Gold-binding properties and kinetics of the native and synthetic CODH-Ls as determined using quartz crystal microbalance and surface plasmon resonance analyses**. The gold-binding activities of the native CODH-L and the synthetic CODH-Ls were examined using quartz crystal microbalance (QCM) analysis, which allows "dynamic" information to be obtained regarding an enzyme adsorption process; this contrasts with the "static" information obtained by AFM. As shown in Fig. 5a, b, the native CODH-L showed negligible binding to the gold surface. This was attributed to the fact that adsorption of the native CODH-L onto the gold surface relies on weak non-specific interactions, which are influenced by the polarity and wettability properties of the surface, in addition to the presence of surface-exposed cysteine residues[56,57].

The CODH-L$_{gbpC}$ system exhibited the strongest gold-binding activity ($\Delta f = 670$ Hz), and this was followed by CODH-L$_{gbpN}$ ($\Delta f = 616$ Hz) and CODH-L$_{gbpNC}$ ($\Delta f = 483$ Hz). In contrast to our initial expectation that the presence of gbp at both ends would give a higher gold-binding activity, the obtained results

indicate that improved binding activities were achieved when the synthetic CODH-L contained gbp at one of its two ends. This unexpected observation can be plausibly explained by considering the immobilization of these enzymes on the electrode. More specifically, in the case of gbp at either the N- or the C-terminus, the enzyme molecules are "standing"; i.e., their longest protein axis is perpendicular to the gold electrode surface because of the site of gbp fusion at the exposed terminus, as confirmed by AFM imaging. Since the enzyme molecules are standing, the calculated enzyme footprint is 52.5 nm². In contrast, in the case of CODH-L$_{gbpNC}$, the presence of two gbp units results in the enzyme molecules "lying down" on the gold surface, i.e., with the longest protein axis parallel to the gold electrode surface. As a result, less space is available because of the larger enzyme footprint, i.e., 63.75 nm². With the experimental value, $\Delta f$, and the molecular weight of the enzymes being known, the surface coverages of the enzymes can be estimated by applying the Sauerbrey equation. The surface coverages of CODH-L, CODH-L$_{gbpN}$, CODH-L$_{gbpC}$, and CODH-L$_{gbpNC}$ were, therefore, determined to be 0.14, 8.19, 8.91, and 6.3 pmol cm$^{-2}$, respectively.

Subsequently, the enzymes were prepared at concentrations of 12.5, 25, 50, 100, and 200 nM; their respective SPR sensorgrams are shown in Fig. 5c, d, while the kinetic adsorption and desorption parameters for all constructs are listed in Table 2. As expected, the results shown in Fig. 5c, d indicate that the native CODH-L exhibits the lowest adsorption rate because of its random non-specific interactions with the gold surface, which render it difficult to undergo stable immobilization, and the enzyme molecules can be easily washed off the surface. The

**Table 2 Binding kinetics of the native and synthetic CODH-Ls as determined by SPR measurements.**

| Analyte | Association rate, $k_a$ (M s$^{-1}$) | Dissociation rate, $k_d$ ($10^{-5}$ s$^{-1}$) | Equilibrium dissociation, $K_D$ (nM) |
|---|---|---|---|
| CODH-L | 1245 | 6.88 | 55.22 |
| CODH-L$_{gbpN}$ | 4983 | 4.08 | 8.19 |
| CODH-L$_{gbpC}$ | 4511 | 12.11 | 26.77 |
| CODH-L$_{gbpNC}$ | 18258 | 3.45 | 1.89 |

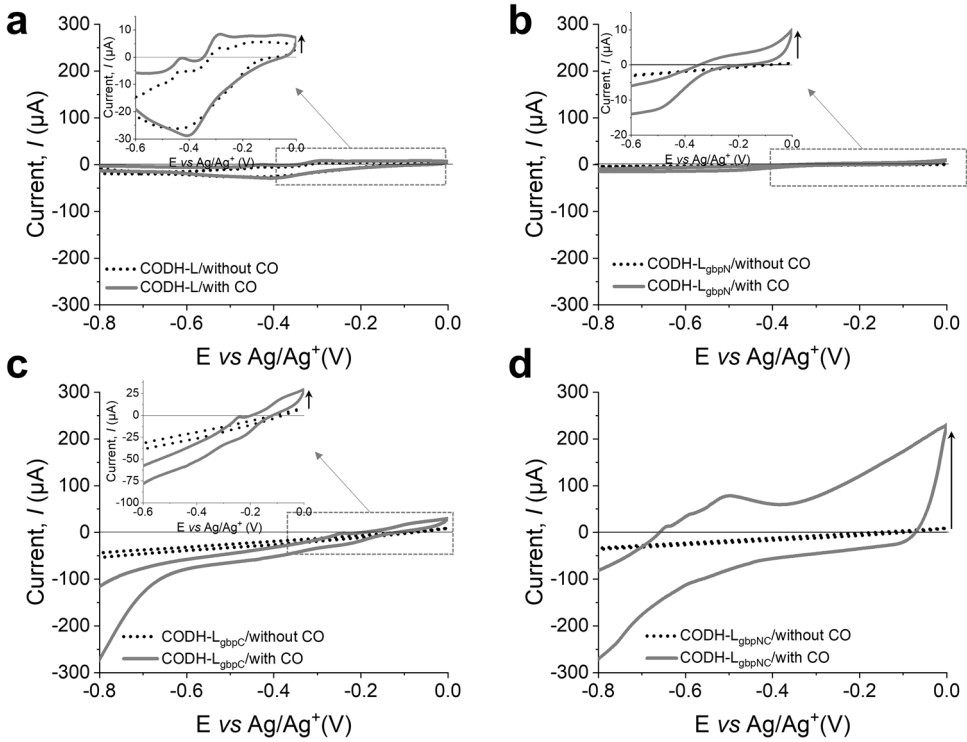

**Fig. 6 Cyclic voltammetry profiles of the native and synthetic enzymes.** Representative CV profiles of **a** the native CODH-L, **b** CODH-L$_{gbpN}$, **c** CODH-L$_{gbpC}$, and **d** CODH-L$_{gbpNC}$ when immobilized on the SPGE in the presence and absence of CO (at a scan rate 100 mV s$^{-1}$).

introduction of the gbp at either the N- or the C-terminus increased the gold-binding ability almost 4-fold over that of the native CODH-L in terms of the association rate constant, $k_a$. However, the dissociation rate constants, $k_d$, of CODH-L$_{gbpN}$ and CODH-L$_{gbpC}$ differed from one another, which was possibly related to the different nearby residues surrounding the gbp-fusion site. Hence, the equilibrium dissociation constant, $K_D$, of CODH-L$_{gbpN}$ was 3-fold smaller than that of CODH-L$_{gbpC}$.

The effect of gbp-fusion at both termini of CODH-L was found to be significant, as evidenced by the dramatic increase in $k_a$ (over 3-fold compared to a single gbp fused at either terminus) and a reduction in the $K_D$ value. Overall, the SPR binding studies demonstrated that fusion of the gbp to the CODH-L did indeed enhance the gold-binding affinity, with the presence of two gbp molecules resulting in a higher adsorption rate and thereby facilitating stable immobilization of the enzyme molecules on the gold surface.

**ET properties at the enzyme–electrode interface.** SPGEs modified with the native and synthetic CODH-Ls were then tested to determine their ET properties using cyclic voltammetry (CV) in phosphate buffer (PB) at pH 7.2. It should be noted that the selection of an appropriate electrolyte for the electrochemical measurements should be based on the assumption that the electrolyte has no effect on the electrochemical characteristics of the

enzymes being studied[58,59]. Thus, 100 mM PB buffer, at pH 7.2 was used to evaluate the bioelectrochemical properties at the enzyme–electrode interface, since this is the optimal buffer composition and pH for this CODH[49]. Fig. 6 shows representative CV profiles for the native and synthetic CODH-Ls in the absence and presence of a CO-saturated PB solution. As a control, CV data were recorded for the bare SPGE both in the presence and absence of CO, and for the enzyme-modified SPGE in the absence of CO. It was found that in the CO-saturated PB solution, all CODH-Ls, with the exception of the CODH-L containing gbp at both termini, exhibited a very weak, almost negligible DET current, indicating that direct electrical contact at the enzyme–electrode interface was not effectively achieved.

A significant DET current was only observed in the case of CODH-L$_{gbpNC}$, for which a pronounced oxidation current onset was observed at $-0.62$ V vs. Ag/Ag$^+$. This onset value agrees with the reported literature, considering that the redox potential of CO oxidation is $-0.76$ V vs. Ag/Ag$^{+60}$. In addition, the CV data show a peak at $-0.5$ V, reflecting the redox potential of Mo–Cu, i.e., $-0.57$ V vs. Ag/Ag$^{+60}$. Although the surface coverage ($\Gamma$, mol cm$^{-2}$) of CODH-L$_{gbpNC}$ was calculated to be 5.38 nmol cm$^{-2}$, indicating the presence of a monolayer on the electrode surface, we observed a cyclic voltammogram with a drawn-out shape. This observation can be explained in terms of the disorder among the immobilized enzyme molecules participating in DET, which results in a spread of

**Table 3 Peak currents, $I_p$, observed for the DET and MET conditions.**

| Enzyme–electrode type | Peak DET current, $I_p$ (µA) | Peak MET current, $I_p$ (µA) | Surface coverage, $\Gamma$, (nmol cm$^{-2}$) | ET rate, $k_{ET}$ (s$^{-1}$) |
|---|---|---|---|---|
| CODH-L | 5.27 ± 3.17 | 105.09 ± 4.95 | N.D. | N.D. |
| CODH-L$_{gbpN}$ | 7.41 ± 2.61 | 334.10 ± 34.11 | N.D. | N.D. |
| CODH-L$_{gbpC}$ | 13.80 ± 6.30 | 332.56 ± 132.56 | N.D. | N.D. |
| CODH-L$_{gbpNC}$ | 195.51 ± 35.51 | 269.02 ± 63.53 | 5.38 | 14.44 |

*N.D. Not determined. Data shown are mean ± s.d for ($n = 3$ independent replicates).

the ET rates, as reflected by the trailing edge in the CV data[40]. More specifically, Leger et al.[40,61] attributed this linear change in current against the driving force to enzyme molecules immobilized on the electrode surface not being orientated in identical configurations, and to this difference in orientation leading to the spread of the interfacial ET rate constants. Thus, the enzyme molecules that are in an ideal orientation contribute to the catalytic signal at low driving forces, while the disorientated enzyme molecules that have poor electrical contact with the electrode can only contribute to the catalytic signal above a certain driving force[40,61]. It should be noted here that CO is the only redox-active species present because the PB does not contain any molecules that produce prominent redox peaks. Supplementary Fig. 5a, c show the dependence of the DET current on the scan rate, wherein the linear dependence of the peak current on the square root of the scan rate indicates that this is a surface-controlled process. In addition, Supplementary Fig. 5c, d show five consecutive scans and chronoamperometry at −0.1 V, respectively. These results reinforce the claim that the oxidation current does originate from the oxidation of CO by CODH at the electrode surface, and that the immobilized enzymes are highly stable.

By introducing the gbp at the N- or the C-terminus of the CODH-L, or at both termini, it was possible to immobilize the different variants at distinct immobilization sites, resulting in a range of ET distances between the active sites and the electrode surfaces. Our results show that efficient DET occurs exclusively when the gbp is fused at both ends of the CODH-L with the kinetic parameter, ET rate ($k_{ET}$), at 14.44 s$^{-1}$, as calculated by plotting the peak potential, $E_p$ against the logarithm of the scan rate, log $\nu$ (Supplementary Fig. 6) based on the Laviron method[62]. This agrees with the estimated distance between the redox center and the electrode surface, i.e., 10.27 ± 1.38 Å (< 14 Å), which theoretically facilitates DET.

To investigate whether the DET catalytic current observed for CODH-L$_{gbpNC}$ was caused by the shorter $d_{ET}$ and to verify whether the difference in the ET kinetics was due to the difference in binding strength, a soluble redox mediator was added to examine the immobilized enzymes for all variants. In this system, the mediator (i.e., methylene blue, MB) should react with all enzyme molecules present at the electrode surface. Thus, the enzyme–electrodes that had been previously tested for DET were tested for MET after the addition of MB (final concentration = 50 µM) to the CO-saturated PB solution. A pronounced mediated oxidation current was observed for all constructs, indicating that the native CODH-L, CODH-L$_{gbpN}$, and CODH-L$_{gbpC}$ were successfully immobilized on the electrode with catalytic activity retention, but that they were not within the necessary DET distance (Supplementary Fig. 7). The oxidation current peak was observed at approximately −0.19 V vs. Ag/Ag$^+$, which is consistent with the standard redox potential of MB at −0.188 V vs. Ag/Ag$^+$. Upon the addition of MB, an ~70 µA increase in the oxidation current was observed for CODH-L$_{gbpNC}$ compared to that observed in the DET current (Table 3). This implies that a fraction of the immobilized enzymes adopted orientations that brought the active sites further from the

electrode surface, or that the enzymes were immobilized in a non-monolayer fashion. As a result, only the first-layer enzyme molecules were able to transfer electrons directly to the electrode surface, and so upon the addition of the MB redox mediator, which can react with all enzymes on the surface, the MET current became significantly higher than the DET current. From these CV results, it was verified that CODH-L$_{gbpNC}$ possessed a direct electrical contact with the electrode surface because there was a suitable ET distance between the two components.

These results rule out the hypothesis that the differences in the ET kinetics observed for the four CODH-L variants are due to differences in the binding strength, as verified from the catalytic current caused by MET, where all constructs were successfully immobilized and retained their biocatalytic activity, but only the synthetic CODH-L$_{gbpNC}$ positioned the active site close to the electrode surface.

## Discussion

Three synthetic CODH-Ls with gbp fused at the N- or the C-terminus, or at both termini, were specifically designed to allow immobilization of the enzyme at various points or sites; the distance between the cofactor and the immobilization site varied accordingly. It was found that the native CODH-L exhibited a low gold-binding affinity as expected, and the enzymes were randomly oriented because of non-specific binding between the native enzyme and the gold surface. When the gbp was fused at either terminus or at both termini, significant increases in the binding affinity toward the gold surface were observed using QCM and SPR analyses. In addition, AFM studies on both an SPGE and a planar gold substrate verified that the native and synthetic CODH-Ls were successfully immobilized, and that CODH-L$_{gbpNC}$ exhibited an almost uniform orientation when bound to the electrode surface. Our experimental results, including electrochemical measurements, confirmed that a pronounced DET occurred exclusively in the case in which the gbp was fused at two sites, since this led to the positioning of the cofactor within the DET distance (i.e., <14 Å) from the electrode surface. In contrast, a very weak DET current (almost negligible) was observed for the native CODH-L, CODH-L$_{gbpN}$, and CODH-L$_{gbpC}$ species. To account for this difference, mediated ET measurements carried out in the presence of MB as a redox mediator showed that the native CODH-L, CODH-L$_{gbpN}$, and CODH-L$_{gbpC}$ were indeed successfully immobilized but that the distance between the redox center and the electrode surface (i.e., the enzyme immobilization point) exceeded 14 Å, reaching 29 and 39 Å for CODH-L$_{gbpN}$ and CODH-L$_{gbpC}$, respectively. Hence, they were unable to undergo DET effectively. It is worth noting that although the surface morphologies of the planar gold surface and the SPGE are different, the electrochemical result revealed that only the CODH-L possessing two tethering sites (i.e., CODH-L$_{gbpNC}$) was available for efficient DET, which is consistent with our distance predictions and AFM results obtained using a planar gold surface. It is therefore suggested that CODH-L$_{gbpNC}$ acquired a stable orientation and that the

**Table 4 Feasibility prediction for the proposed platform technology for application with various enzymes commonly used in the enzyme bioelectrocatalytic system based on our system criteria.**

| Enzyme types | Organism source | Protein data bank (PDB) ID | 1) Are both N- and C- termini solvent-exposed? | 2) Are both N- and C- termini non-contacted with catalytic domain? | 3) Estimated ET distance, $d$ (Å) | Possible DET via dual tethering sites at both termini? |
|---|---|---|---|---|---|---|
| Bilirubin oxidase | Magnaporthe oryzae | 6IQZ | Yes | Yes | 13.06 ± 2.95 | Yes |
| Cellobiose dehydrogenase | Myriococcum thermophilum | 4QI6 | No | No | N.D.[a] | No |
| FAD-dependent glucose dehydrogenase | Burkholderia cepacia | 6A2U | Yes | Yes | 10.84 ± 0.16 | Yes |
| Formate dehydrogenase | Candida boidinii | 5DNA | Yes | Yes | 12.37 ± 0.40 | Yes |
| Glucose oxidase | Apergillus niger | 1CF3 | Yes | Yes | 13.93 ± 0.64 | Yes |
| Hydrogenase | Ralstonia eutropha | 3RGW | No | No | N.D.[a] | No |
| Laccase | Trametes versicolor | 1KYA | No | No | N.D.[a] | No |
| Multicopper oxidase | Pyrobaculum aerophilum | 3AW5 | No | No | N.D.[a] | No |
| NAD-dependent alcohol dehydrogenase | Saccharomyces cerevisiae | 1PS0 | Yes | Yes | N.D.[a] | No |
| NAD-dependent glucose dehydrogenase | Bacillus megaterium | 3AUS | Yes | Yes | 13.80 ± 2.94 | Yes |
| PQQ-dependent glucose dehydrogenase | Acinetobacter calcoaceticus | 5 min | Yes | Yes | 13.60 ± 2.95 | Yes |
| Tungsten-containing formate dehydrogenase | Desulfovibrio gigas | 1H0H | Yes | Yes | 12.37 ± 4.07 | Yes |

[a]Could not be determined because the terminus either is not solvent-exposed or has contact with the catalytic domain (i.e., does not satisfy criteria 1 and 2). Data shown are mean ± s.d for 3 shortest ET distance (i.e., distance between cofactor and solid support).

orientation of CODH-L possessing gbp at either the N- or the C-terminus would be flexible due to the high specificity of gbp toward the gold surface, either in an atomically flat or structured topography. Therefore, our estimation regarding the enzyme orientation and the ET distance do not apply to only atomically flat surfaces, but also to the SPGE surface.

Using LOMETS, the enzyme structures, and PyMol analysis, we subsequently examined the feasibility of employing our proposed platform technology on enzymes that are commonly employed in bioelectrocatalytic systems, as tabulated in Table 4. We considered that this approach, in which the structure-guided fusion site selection for two tethering points resulted in direct electrical connection, should be readily adaptable for other redox enzymes regardless of the type of fusion or the solid binding peptide employed; however, it is exclusively applicable to enzymes with 3D structures in which the N- and C- termini are freely exposed to the bulk solvent and have no contact with the catalytic domain, and in which the enzyme cofactor-surface distance is <14 Å, as summarized in Table 4. It must be emphasized that interfacial DET is governed by many factors not limited to the ET distance, and often there are too many variables to arrive at a firm conclusion, especially in the context of complex biotic-abiotic systems, such as enzyme–electrode systems. Importantly, we demonstrated that the site-directed immobilization of an enzyme via two tethering sites enabled the enzyme to adopt a specific controlled orientation, which could be adapted to other immobilization techniques, including but not limited to, covalent coupling, click-chemistry based, affinity-based, and cysteine-dependent immobilization on gold electrodes. Controlling the enzyme orientation is not only critical in the context of interfacial DET in a single enzyme–electrode reaction, but also an enzyme cascade reaction where multiple enzymes should ideally adopt specific orientations for facile substrate (intermediate) channeling. Moreover, controlled enzyme orientation on an electrode surface is also important in the context of MET because the enzyme molecules should be oriented in a manner that renders the redox site accessible to both the substrate and a small redox mediator for fast ET. More generally, this approach to structure-guided protein engineering for site-controlled immobilization is expected to find broad applications, including enzyme-electrode systems, biochips, protein-protein (or ligand) interactions, drug screening, and single-molecule analysis.

## Methods

**Recombinant production of CODH-L and its variants**. Plasmid pET21a (+), harboring the *cutL* gene encoding the L-subunit of CODH from *H. pseudoflava*, was successfully constructed and expressed as described previously[49]. This plasmid was used as the template for the construction of synthetic CODH-L fused with LKAHLPPSRLPS at various sites of the CODH-L. LOMETS was employed to generate a contact and distance map to carefully select specific site(s) in the protein for gbp fusion. From the LOMETS results, only the N- (residue 1) and the C-terminus (residue 803) were identified as possible sites for gbp fusion because these two termini have no contact with the main domain. More specifically, from the 3D structure (PDB ID: 1FFV), both termini were solvent-exposed and faced away from the main domain. Thus, the gbp was fused at the desired site(s) via site-directed mutagenesis using the primers listed in Supplementary Table 3 and following the NEBaseChanger protocol using a Q5 Site-Directed Mutagenesis Kit from New England Biolabs (NEB, MA, USA). To separate the active site domain from the gbp-binding domain, 5× glycine was used as a flexible linker between the CODH and the gbp. Supplementary Table 4 lists all strains and plasmids used in this study. All plasmids were extracted, purified from DH5α, sequenced, and verified by Solgent Co. (Seoul, Korea) prior to their transformation into *E. coli* BL21-AI. The full amino acid sequences of the native and synthetic CODH-Ls used in this study are available in Supplementary Table 5.

For overexpression in *E. coli* BL21-AI, each expression combination was grown in 2×YT medium containing 100 μg mL$^{-1}$ ampicillin at 37 °C and 200 rpm. When the optical density, OD$_{600}$, reached ~0.5, IPTG (1 mM), Na$_2$MoO$_4$ (1 mM), and CuSO$_4$ (0.1 mM) were added for induction and in vivo reconstitution. The temperature was then lowered to 16 °C. After cultivation for 24 h, the cells were harvested by centrifugation (4000 rpm, 10 min, 4 °C). The cell pellet was then

resuspended in a resuspension buffer containing 50 mM sodium PB at pH 8, 300 mM NaCl, 10 mM imidazole, 10% glycerol, and an ethylenediaminetetraacetic acid-free protease inhibitor (Roche, Switzerland). Unless otherwise stated, all steps were conducted at 4 °C. Subsequently, the cells were disrupted by ultrasonication over three cycles of 3 min, and the disrupted cells were subjected to centrifugation at 7000 rpm for 30 min, after which the supernatant was filtered using a 0.45 μm membrane filter. The filtered supernatant was loaded onto a column packed with cobalt resin and then washed with 50× washing buffer containing 50 mM sodium phosphate (pH 7.5), 300 mM NaCl, 20 mM imidazole, and 10% glycerol. Elution was then performed using 100 mM potassium PB (pH 7.2) containing 300 mM NaCl, 20 mM imidazole, and 10% glycerol. The eluted fractions containing the target protein were then pooled and buffer exchanged to 100 mM PB (pH 7.2) and 10% glycerol, then finally concentrated using an Amicon filter (30 kDa cutoff).

**Enzyme activity assay**. The CO oxidation activities of the CODHs were measured in solution, based on the method employed by Zhang et al.[63] with slight modifications. More specifically, nitrogen was bubbled through the reaction mixture containing PB (0.1 M, pH 7.2) and methylene blue (20 μM) for 15 min to ensure deoxygenation of the solution. Pure CO was then bubbled through the solution for at least 30 min to achieve a dissolved CO concentration of 1 mM, and the CO-saturated reaction mixture was preheated to 30 °C prior to analysis. For analysis, an aliquot (0.9 mL) of the reaction mixture was transferred carefully using a gas-tight syringe into a gas-tight rubber-capped cuvette purged with nitrogen. The enzyme activity was measured by adding an aliquot (100 μL) of the anaerobically prepared enzyme solution, and the reduction of methylene blue was followed spectro-photometrically at 615 nm using a UV spectrophotometer (TS Science, Seoul, South Korea). The enzyme activity was then calculated using the Beer–Lambert law, as described previously[49]. One unit (U) is defined as the oxidation of 1 μmol of CO per minute under the given conditions.

**Enzyme immobilization on the electrode**. A SPGE (250 AT) from Dropsens (Spain) comprising a gold working electrode (4 mm diameter), a silver pseudor-eference electrode (Ag/Ag$^+$), and a platinum counter electrode was used. The working electrode of the SPGE was washed with deionized water three times for 20 s prior to each use. Subsequently, the SPGE was conditioned with PB and dried using a blower. The protein concentration was then adjusted to 1 μM, and an aliquot (700 μL) of each protein sample was incubated with the SPGEs at 16 °C for 2 h. After this time, the surfaces were rinsed with deionized water for 10 s and dried with a blower. The freshly prepared enzyme-immobilized SPGEs were then used directly for each measurement.

For immobilization on the planar gold substrates, the gold substrates (1 cm × 1 cm) were cleaned with a mixture of piranha solution and 30% H$_2$O$_2$/H$_2$SO$_4$ (1:3 (v/v)) for 20 min and then rinsed with deionized water. Subsequently, an aliquot (500 μL) of each protein sample (0.5 μM or 1 μM) was incubated with a gold substrate for 2 h at 16 °C, after which time the surfaces were washed with deionized water for 10 s and gently dried using a blower.

**AFM measurement**. The surfaces were imaged using the non-contact tapping mode (XE-100, Park Systems, Langen, Germany). Measurement was carried out using a scan rate of 0.3 Hz with a scan size of 5 × 5 μm. Surface analysis was performed with XEI-software (Park Systems).

**QCM analysis**. The qualitative binding properties of the enzymes were measured using QCM (QCM922A SEIKO EG & G, Tokyo, Japan). The gold-coated chips of the QCM sensor were obtained from the same manufacturer. In real-time adsorption measurements, a stable baseline was obtained by allowing a sufficient volume (i.e., >5 mL) of the 1 M phosphate running buffer at pH 7.2 to flow into the cell at a flow rate of 143 μL min$^{-1}$ for 30–60 min. When a stable baseline was obtained, the protein sample (1 μM, 1.5 mL) was added, and the frequency change was recorded continuously to monitor the adsorption process. Following sample addition to the flow, a portion (3–4 mL) of running buffer was added, and the desorption process was observed continuously via QCM analysis.

**SPR measurements**. The gold-binding kinetics of the enzymes were measured by SPR (SR7500DC, Reichert Analytical Instruments, Depew, NY) using a bare gold chip (RE13206060, Reichert, Depew, NY). The kinetic titration model used in this study was for a five-injection kinetic titration without mass transport. The association and dissociation times were set at 3 and 5 min, respectively. The data were analyzed using Scrubber 2.0 and ClampXP. For data analysis, the 1:1 interaction model was used to calculate the association rate constant ($k_a$) and the dissociation rate constant ($k_d$), while the equilibrium ($K_D$) constant was calculated as the ratio of $k_d/k_a$ determined from the kinetic experiments[43].

**Electrochemical measurements**. A gas-tight reactor (working volume = 8 mL) equipped with a magnetic stirrer bar was used to maintain the CO-saturated solution for the electrochemical tests. The Dropsens CAC interface was connected to a potentiostat (Autolab, Metrohm AutoLab, Utrecht, Netherlands). Prior to carrying out the electrochemical measurements, the electrolyte was saturated with

CO (99.95%, Daedeok Gas, Korea) for 30 min, after which time, no significant pH change was observed. All electrochemical tests were conducted in a temperature-controlled room at 30 °C, unless otherwise stated.

**Statistics and reproducibility**. All data analysis was performed using Excel or Origin software. Reproducibility was confirmed by performing three independent replicates as stated in the figure legend.

**Reporting summary**. Further information on research design is available in the Nature Research Reporting Summary linked to this article.

## Data availability
Source data for underlying the graphs and plots in the main figures are provided in Supplementary Data. The uncropped image of SDS-PAGE gel in Fig. 1d is included in the Supplementary Information file as Supplementary Fig. 1. Any remaining information can be obtained from the corresponding author upon reasonable request.

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

## Acknowledgements

This work was supported by grants from the National Research Foundation of Korea (NRF), funded by the Korean Government (no. 2020R1A2C3009210 and no. 2021R1A5A1028138). The authors thank Professor Jun-Ho Choi of the Department of Chemistry, at the Gwangju Institute of Science and Technology, Korea for his advice regarding protein structures. The authors also thank Professor Jae Young Lee of the School of Materials Science and Engineering, Gwangju Institute of Science and Technology, Korea for providing the QCM instrument.

## Author contributions

S.S.R. proposed the concept and performed the experiments and data analysis. S.S.R. and H.L. performed the QCM experiments. S.S.R., N.F., and B.S. performed the protein structure analysis and animation. M.L. assisted in enzyme assay experiments. M.J.K assisted in protein production. S.S.R., H.B., and I.S.C. wrote and edited the manuscript. I.S.C. conceived the project and supervised the research. The final manuscript was approved by all authors.

## Competing interests

The authors declare no competing interests.
