## [Peer Review File · Communications Biology]

Reviewers' comments:

Reviewer #1 (Remarks to the Author):

The paper by Reginal et al. describes the orientation of carbon monoxide dehydrogenase (CODH) to enhance DET on electrode surface. While the paper is interesting it may lack of novelty compared to their previous work on the same topic. Publication must be considered prior major revisions, below some points:

- Many different techniques have been used and site-specific immobilization is designed and explained in detail. However it is too specific for CODH orientation purpose, would the same principle and method work for common enzymes i.e. laccase, gox, bod etc.? Please discuss
- At page 4 line 84 a model enzyme was designed and engineered to fuse gpb at specific sites. How is it possible to consider it as model enzyme if it is specifically engineered?
- PB buffer 100 mM was used to evaluate enzyme activity. Supporting electrolytes are known to affect redox enzymes performances (i.e. 10.1016/j.coelec.2017.08.011, 10.1039/C4CS00144C) how does the this electrolyte affect the DET? Could the catalytic activity be enhanced or tuned by PB or another different electrolyte? Please add some reference and discuss
- References are not on spot, they can be fulfilled with more recent works on bioelectrocatalytic activity, electrodes design for DET improvements and so forth.
- English must be carefully revised.

Reviewer #2 (Remarks to the Author):

The authors have conducted a nicely organized and well thought out study of the effect of adding one (either C-terminus or N-terminus)gold binding peptide or two (both C-terminus and N-terminus)to the enzyme carbon monoxide dehydrogenase determines whether or not direct electron transfer occurs. The authors have made a good effort to estimate the distance between the enzyme active site and the gold surface and show that the form with two GBPs comes withing the required distance and shows clear catalytic activity. The supplementary information is informative with additional EDS, electrochemical and other information. Table IV is especially valuable suggesting for which other enzymes the strategy may work. The reported results seem quite likely to inspire further studies to enhance DET.

Some questions to consider are:

- (1) The amounts of Mo and Cu do increase significantly upon immobilization in EDS data. Is a rough estimate of comparisons of coverages for the four possibilities consistent with the other results or is the initial amount of Mo and Cu on the surface too variable?
- (2) For Figure 2, is it possible to fit to an adsorption isotherm and predict a saturation coverage?
- (3) Can the mechanism of gold binding by unmodified enzyme be discussed in a bit more detail
- (4) Is it possible to show in SI a Laviron plot for determination of the electron transfer rate constant?

Reviewer #3 (Remarks to the Author):

The submitted work is dedicated to the investigation of the orientation of the enzyme on the gold supporting electrode and impact of its consequence onto its activity towards CO. Authors carried out experimental work focused onto the electrode preparation and the modification by the enzyme, morphology and composition analysis of the obtained structures using AFM, SEM coupled with EDS. There were also carried out electrochemical measurements in the presence and absence CO in the electrolyte enabling analysis of the electrical contact between the current collector and the enzyme. In ESI authors provides also very nice animation of the protein immobilization. Finally, authors promises platform technology ensuring enzyme orientation with side specific orientation.

The work is clearly written, using simple language and adequate terminology.

Because of my main research experience area, given below comments and questions will be addressed to the electrochemical part and the investigations of the morphology and composition:

a) What is the orientation of the gold surface in the case of planar gold surface and screen printed gold electrode. Authors are asked to verify it and explain the possible effect onto the surface modification and further electrode electrochemical activity.

b) Fig. 4 – AFM of unmodified gold surface should be included (at least in ESI file) since described difference in topography is very debatable.

c) In ESI-Fig.4 authors include SEM coupled with EDS showing the presence of Mo and Cu also for the screen printed gold electrode – why both molybdenum and cobalt is present in that electrode despite any enzyme should not be on the surface? Especially the signal originating from Mo is clearly visible for SPGE.

d) In my opinion authors are too enthusiastic about regarding the difference in the content of Mo and Cu and this analysis is highly debatable since still it is unclear why Mo is present in SPGE.

e) How the surface coverage was calculated?

f) The experimental part regarding the electrochemical measurements should be enriched by more information:

- How long the electrolyte was saturated with CO?
- How pH of the electrolyte change after CO saturation?
- What is the electrolyte temperature (the temperature conditions were only provided for protein immobilization occurring @16 deg C) and did authors observe any change if the temperature was not controlled?
- Did CO flow above the electrolyte was maintained during electrochemical measurements?
- What is the purpose of conditioning the electrode surface in PB?
- What is the purity of CO gas used for saturation?

g) In ESI-Fig. 5b – the unit of the square root of scanning speed is missing and at which potential the peak current was taken to prepare Fig ESI-5b?

h) What was the final concentration of MB in PB solution since 50 micromol MB was added. Such description indicates that the initial concentration of MB equals 50 micromol but the final one can significantly differ depending of the volume of the electrolyte.

i) What is the repeatability of the electrochemical activity for gold electrode modified with an enzyme?

j) In one part of the "Methods" authors write about PB with pH=8 and in the other that pH=7.2. How pH of PBS was changed or it is just mistake in some of those values?

k) Authors described the significance of the obtained results as important for the fabrication of DET-based bioelectrocatalysis system. Can authors develop more the importance of their results?

l) Can author clearly indicate why electrochemical tests were not carried out for the modified bare gold disc electrode, since in my opinion the justification is unclear and not well explained?

I hope that those comments will be helpful in the correction of those projects?

Reviewer's comments

< Reviewer 1 >

The paper by Reginal et al. describes the orientation of carbon monoxide dehydrogenase (CODH) to enhance DET on electrode surface. While the paper is interesting it may lack of novelty compared to their previous work on the same topic. Publication must be considered prior major revisions, below some points:

1. Many different techniques have been used and site-specific immobilization is designed and explained in detail. However it is too specific for CODH orientation purpose, would the same principle and method work for common enzymes i.e. laccase, gox, bod etc.? Please discuss

Thank you for this comment and question. Considering that enzymes are complex biomacromolecules having diverse functions and structures, we examined the possibility of employing the same method to other common enzymes used in bioelectrocatalytic systems, as indicated in Table 4. We expect that this principle and method may not be applicable to all enzymes, but rather only to enzymes that possess specific features. For example, the fusion site (i.e., the N- and C-termini) should be freely exposed to the bulk solvent, the resulting gold-binding domain should have no contact with the main (catalytic) domain, and the active site or cofactor should be within the DET-available distance (i.e., $<14 \text{ \AA}$) from the electrode surface. For instance, due to the fact that glucose oxidase and bilirubin oxidase both possess these features, our approach would be expected to result in DET; however, in the case of laccase, our method would likely not be applicable as it does not fulfill the required criteria. It must be emphasized that interfacial DET is governed by many factors not limited to the ET distance, and often there are too many variables to arrive at a firm conclusion, especially in the case of hybrid systems, such as the enzyme–electrode system. Importantly, we showed that the site-directed immobilization of an enzyme via two tethering sites enabled the enzyme to adopt a specific controlled orientation, which could be adapted to other immobilization techniques, including but not limited to, covalent coupling, click-chemistry based, affinity-based, and cysteine-dependent immobilization on gold electrodes. To reflect reviewer's comment, the following text has now been added to the manuscript:

Page 19, line 394–401:

“It must be emphasized that interfacial DET is governed by many factors not limited to the ET distance, and often there are too many variables to arrive at a firm conclusion, especially in the

context of complex biotic-abiotic systems, such as enzyme–electrode system. Importantly, we demonstrated that the site-directed immobilization of an enzyme via two tethering sites enabled the enzyme to adopt a specific controlled orientation, which could be adapted to other immobilization techniques, including but not limited to, covalent coupling, click-chemistry based, affinity-based, and cysteine-dependent immobilization on gold electrodes.”

2. At page 4 line 84 a model enzyme was designed and engineered to fuse gpb at specific sites.

How is it possible to consider it as model enzyme if it is specifically engineered?

We thank the reviewer for this question. In the sentence that the reviewer mentions, we are referring to the unmodified CODH as the model enzyme for our study. For clarity purposes, we have now revised this sentence by deleting the word “model,” as follows:

Page 4, Lines 83–85:

“For this purpose, an enzyme was designed and engineered to fuse the gpb at specific sites, to generate various immobilization sites, and to control the distance between the immobilization site and the cofactor (i.e., the only variable parameter to consider).”

3. PB buffer 100 mM was used to evaluate enzyme activity. Supporting electrolytes are known to affect redox enzymes performances (i.e. 10.1016/j.coelec.2017.08.011, 10.1039/C4CS00144C) how does the this electrolyte affect the DET? Could the catalytic activity be enhanced or tuned by PB or another different electrolyte? Please add some reference and discuss.

We thank the reviewer for the comment and helpful references. During our efforts to purify this CODH, we tested several buffer compositions (i.e., Tris-HCl of various pH and compositions, and phosphate buffer of various pH and compositions), but the protein became insoluble, indicating its sensitivity to changes in the buffer composition. Therefore, the choice of the electrolyte for the electrochemical measurements was taken based on our previous result that CODH exhibited its optimal activity in this electrolyte, and the assumption that the electrolyte itself does not affect the bioelectrochemical properties of CODH. Thus, 100 mM PB buffer at pH 7.2 was used to evaluate the enzyme and bioelectrocatalytic activity as this was considered the optimal buffer composition and pH for this CODH. We considered that it is unlikely that the use of an alternative electrolyte would enhance the catalytic activity and the DET process, at least in the case of this CODH due to its sensitivity to the change in buffer compositions. The manuscript has now been revised according to justify our choice of electrolyte for the electrochemical measurements.

Page 14, line 291–295:

“It should be noted that the selection of an appropriate electrolyte for the electrochemical measurements should be based on the assumption that the electrolyte has no effect on the electrochemical characteristics of the enzymes being studied. Thus, 100 mM PB buffer, at pH 7.2 was used to evaluate the bioelectrochemical properties at the enzyme–electrode interface, since this is the optimal buffer composition and pH for this CODH.”

4. References are not on spot, they can be fulfilled with more recent works on bioelectrocatalytic activity, electrodes design for DET improvements and so forth.

We thank the reviewer for this suggestion. A number of more recent references in relation to DET improvements in enzyme–electrode systems have now also been included as follows:

References

29. Han, Z., Zhao, L., Yu, P., Chen, J., Wu, F., Mao, L. Comparative investigation of small laccase immobilized on carbon nanomaterials for direct bioelectrocatalysis of oxygen reduction. *Electrochem. Commun.* **101**, 82-87 (2019).
30. Kizling, M., Dzwonek, M., Więckowska, A., Bilewicz, R. Size does matter—Mediation of electron transfer by gold clusters in bioelectrocatalysis. *ChemCatChem.* **10**, 1988-1992 (2018).
34. Yan, X. *et al.* Direct electron transfer of fructose dehydrogenase immobilized on thiol-gold electrodes. *Electrochim. Acta* **392**, 138946 (2021).
35. Badiani, V.M. *et al.* Elucidating film loss and the role of hydrogen bonding of adsorbed redox enzymes by electrochemical quartz crystal microbalance analysis. *ACS Catal.* **12**, 1886-1897 (2022).
36. Algov, I., Grushka, J., Zarivach, R., Alfonta, L. Highly efficient flavin–adenine dinucleotide glucose dehydrogenase fused to a minimal cytochrome C domain. *J. Am. Chem. Soc.* **139**, 17217-17220 (2017).
37. Yanase, T., Okuda-Shimazaki, J., Mori, K., Kojima, K., Tsugawa, W., Sode, K. Creation of a novel DET type FAD glucose dehydrogenase harboring Escherichia coli derived cytochrome b(562) as an electron transfer domain. *Biochem. Biophys. Res. Commun.* **530**, 82-86 (2020).
38. Ito, K. *et al.* Designer fungus FAD glucose dehydrogenase capable of direct electron transfer. *Biosens. Bioelectron.* **123**, 114-123 (2019).
39. Viehauser, M.-C., Breslmayr, E., Scheiblbrandner, S., Schachinger, F., Ma, S., Ludwig, R.

A cytochrome b-glucose dehydrogenase chimeric enzyme capable of direct electron transfer. *Biosens. Bioelectron.* **196**, 113704 (2022).

45. Algov, I., Feiertag, A., Alfonta, L. Site-specifically wired and oriented glucose dehydrogenase fused to a minimal cytochrome with high glucose sensing sensitivity. *Biosens. Bioelectron.* **180**, 113117 (2021).

5. English must be carefully revised.

We thank the reviewer for this comment. This manuscript has now been grammatically corrected and proof-read by a professional English Language Editing Service (Editage) prior to submission.

< Reviewer 2 >

The authors have conducted a nicely organized and well thought out study of the effect of adding one (either C-terminus or N-terminus) gold binding peptide or two (both C-terminus and N-terminus) to the enzyme carbon monoxide dehydrogenase determines whether or not direct electron transfer occurs. The authors have made a good effort to estimate the distance between the enzyme active site and the gold surface and show that the form with two GBPs comes within the required distance and shows clear catalytic activity. The supplementary information is informative with additional EDS, electrochemical and other information. Table IV is especially valuable suggesting for which other enzymes the strategy may work. The reported results seem quite likely to inspire further studies to enhance DET.

Some questions to consider are:

1. The amounts of Mo and Cu do increase significantly upon immobilization in EDS data. Is a rough estimate of comparisons of coverages for the four possibilities consistent with the other results or is the initial amount of Mo and Cu on the surface too variable?

We thank the reviewer for this comment and question. Although our interpretation is straightforward in terms of the SEM/EDS analysis, it is not completely clear why Mo and Cu were present on the bare SPGE surface, as this is a commercially available SPGE. We are therefore unable to clarify this point. In addition, we note that considering the fact that CODH-L contains 2 moles of Mo and Cu per enzyme subunit, it would be difficult to observe any noticeable differences in terms of Mo and Cu based on elemental mapping analysis since the signal originating from Mo is clearly visible for SPGE. To reflect the reviewer's comment, we have omitted the SEM/EDS results since they do not provide any meaningful data for clear interpretation.

2. For Figure 2, is it possible to fit to an adsorption isotherm and predict a saturation coverage?

We thank the reviewer for this suggestion, and we note that we have assumed that the reviewer is referring to Figure 5. Thus, in this context, the gold-binding kinetics of the enzymes were determined using the kinetic titration method (also known as the single-cycle kinetics (SCK) method), which consists of sequential injections of increasing concentrations of the enzyme without any regeneration steps between each sample injection. This method is considered more efficient

than the conventional method (alternating cycles of analyte injections and surface regeneration) for characterizing analyte–ligand binding reactions that are difficult to regenerate. The kinetic parameters used in this study were extracted using the kinetic 1:1 binding model. We note that one limitation of this single-cycle kinetics method is the reduced informational content obtained from the single dissociation phase of an interaction under study. As such, we are not able to predict the saturation coverage, which can be calculated using the De Feijter equation (<https://doi.org/10.1002/bip.1978.360170711>, [10.3390/bios11060180](https://doi.org/10.3390/bios11060180)). Thus, to reflect reviewer’s comment, we have revised Figure 5c to include blue background traces to represent the global fit of the data to a 1:1 binding model. The titration curves for the various CODH-L species, which are depicted in Figure 5d, were generated from the raw experimental data shown in Figure 5c by plotting the changes in the binding response against the enzyme concentration.

Figures 5c and 5d

c) Changes in the SPR refractive index as a function of time for enzyme adsorption on the gold surface at enzyme concentrations of 12.5, 25, 50, 100, and 200 nM (the solid arrows indicate enzyme injection while the dashed arrows indicate buffer injection). **The blue traces represent the global fit of the data to a 1:1 binding model.**

3. Can the mechanism of gold binding by unmodified enzyme be discussed in a bit more detail

We thank the reviewer for this suggestion. We note that the non-specific binding between the unmodified enzyme (enzyme without gbp) and the gold surface is influenced by a number of factors, including the polarity, charge, wettability properties, and surface-exposed cysteine residues ([10.1021/acs.langmuir.1c01033](https://doi.org/10.1021/acs.langmuir.1c01033), [10.1021/acsomega.1c03774](https://doi.org/10.1021/acsomega.1c03774)). However, these non-specific interactions between the enzyme and the gold surface are particularly weak, and so the enzyme can be easily desorbed from the gold surface, as evidenced our the SPR and QCM results. In contrast, the enzyme fused with gbp exhibits an enhanced gold-binding activity. Although it is well

understood that the interactions between a solid-binding peptide and a target solid material are based on the chemical groups present within the amino acid residues having a high affinity for the solid surface, the presence of such specific amino acids did not always translated into an improved binding affinity for an entire peptide to a specific material. This can be attributed to the fact that the exact amino acid sequence is considered more important than the composition in determining the function of the solid-binding peptide (10.1016/j.tibtech.2015.02.005). At this point, the mechanism responsible for gbp recognition, the enhanced binding affinity, and the observed selectivity remains poorly understood due to the complexity of the peptide-solid material interactions. To reflect reviewer's comment, we have added the following sentence to the revised text:

Page 12, lines 241–244:

“This was attributed to the fact that adsorption of the native CODH-L onto the gold surface relies on weak non-specific interactions, which are influenced by the polarity and wettability properties of the surface, in addition to the presence of surface-exposed cysteine residues.”

4. Is it possible to show in SI a Laviron plot for determination of the electron transfer rate constant? We thank the reviewer for this suggestion. Accordingly, to determine the electron transfer rate constant, a Laviron plot of the peak potential (E_p) against the logarithm of the scan rate ($\log \nu$) has been added to the Supplementary Information as Supplementary Figure 5. To reflect reviewer's comment, we have added the following sentence to the revised text:

Page 17, lines 332–333:

... as calculated by plotting the peak potential, E_p against the logarithm of the scan rate, $\log \nu$ (Supplementary Fig. 5) based on the Laviron method.

Supplementary Figure 5

Supplementary Figure 5. Variation of peak potential, E_p against the logarithm of scan rates, $\log v$.

< Reviewer 3 >

The submitted work is dedicated to the investigation of the orientation of the enzyme on the gold supporting electrode and impact of its consequence onto its activity towards CO. Authors carried out experimental work focused onto the electrode preparation and the modification by the enzyme, morphology and composition analysis of the obtained structures using AFM, SEM coupled with EDS. There were also carried out electrochemical measurements in the presence and absence CO in the electrolyte enabling analysis of the electrical contact between the current collector and the enzyme. In ESI authors provides also very nice animation of the protein immobilization. Finally, authors promises platform technology ensuring enzyme orientation with side specific orientation.

The work is clearly written, using simple language and adequate terminology.

Because of my main research experience area, given below comments and questions will be addressed to the electrochemical part and the investigations of the morphology and composition:

1. What is the orientation of the gold surface in the case of planar gold surface and screen-printed gold electrode. Authors are asked to verify it and explain the possible effect onto the surface modification and further electrode electrochemical activity.

We thank the reviewer for their good understanding of our manuscript and for the helpful comments. In terms of this first point, we note that the planar gold surface with an atomically flat surface (Supplementary Figure 1a) allowed imaging to be carried out using AFM, and therefore it was reassuring to obtain a clear visual of the enzyme immobilized on this atomically flat gold surface. The AFM result therefore provides us an excellent guide as to what orientation can be expected for the four types of CODH constructs when immobilized on an atomically flat gold surface. As Armstrong previously described (<https://doi.org/10.1016/j.electacta.2021.138836>), in reality, practical electrode surfaces are typically not flat, such in the case of the screen-printed gold electrode used in this study (Supplementary Figure 3a). Since the bioelectrocatalytic properties of the enzyme-modified electrode were found to be consistent with our distance prediction (i.e., only the CODH bearing gbp at both termini exhibited excellent DET due to the shorter ET distance), the predicted enzyme orientation is likely not limited to the atomically flat surface, but could also apply to a smooth SPGE surface containing some voids and raised regions on the micrometer scale, in which each enzyme is considered to be in its own local environment. Additionally, it must be emphasized that enzyme molecules are dynamic, and so it is rational to expect that the CODH without or with a single tethering site would have the freedom to explore various orientations on the gold surface. In contrast, the CODH bearing the gbp at both termini should acquire a stable

orientation regardless of the surface properties due to the presence of the two tethering sites.

To reflect these points, the discussion has been updated as follows:

Page 19, lines 378–386:

“It is worth noting that although the surface morphologies of the planar gold surface and the SPGE are different, the electrochemical result revealed that only the CODH-L possessing two tethering sites (i.e., CODH-L_{gbpNC}) was available for efficient DET, which is consistent with our distance predictions and AFM results obtained using a planar gold surface. It is therefore suggested that CODH-L_{gbpNC} acquired a stable orientation, and that the orientation of CODH-L possessing gbp at either the N- or the C- terminus would be flexible due to the highly specificity of gbp toward the gold surface, either in an atomically flat or structured topography. Therefore, our estimation regarding the enzyme orientation and the ET distance do not apply to only atomically flat surfaces, but also to the SPGE surface.”

2. Fig. 4 – AFM of unmodified gold surface should be included (at least in ESI file) since described difference in topography is very debatable.

We thank the reviewer for this suggestion. Accordingly, the AFM results for the unmodified gold surface have now been provided in the ESI as Supplementary Figure 1(a).

Supplementary Figure 1(a)

3. In ESI-Fig.4 authors include SEM coupled with EDS showing the presence of Mo and Cu also for the screen printed gold electrode – why both molybdenum and cobalt is present in that electrode despite any enzyme should not be on the surface? Especially the signal originating from Mo is clearly visible for SPGE.

We thank the reviewer for this comment. Although our interpretation is straightforward in terms of the SEM/EDS analysis, it is not completely clear why Mo and Cu were present on the bare SPGE surface, as this is a commercially available SPGE. We are therefore unable to clarify this point. In addition, we note that considering the fact that CODH-L contains 2 moles of Mo and Cu per enzyme subunit, it would be difficult to observe any noticeable differences in terms of Mo and Cu based on elemental mapping analysis since the signal originating from Mo is clearly visible for SPGE. To reflect the reviewer's comment, we have omitted the SEM/EDS results since they do not provide any meaningful data for clear interpretation.

4. In my opinion authors are too enthusiastic about regarding the difference in the content of Mo and Cu and this analysis is highly debatable since still it is unclear why Mo is present in SPGE. We thank the reviewer for the comment. As the reviewer argues, we do agree that this analysis could be debatable. To reflect reviewer's comment, we have omitted the SEM/EDS results since they do not provide any meaningful data for clear interpretation.

5. How the surface coverage was calculated?

We thank the reviewer for this question, and we note that the surface coverage was estimated using the following equation:

$$i_p = \frac{n^2 F^2}{4RT} \nu A \Gamma$$

where

i_p : peak current

F : Faraday constant

n : number of electrons transferred

R : ideal gas constant

ν : scan rate (Vs^{-1})

A : electrode area (cm^2)

Γ : surface coverage (molcm^{-2})

Taking $n = 2$, $A = 0.12566 \text{ cm}^{-2}$, and $\nu = 0.1 \text{ Vs}^{-1}$. Therefore, the surface coverage was calculated to be $5.385 \text{ nmol cm}^{-2}$

6. The experimental part regarding the electrochemical measurements should be enriched by more information:

- How long the electrolyte was saturated with CO?
- How pH of the electrolyte change after CO saturation?
- What is the electrolyte temperature (the temperature conditions were only provided for protein immobilization occurring @16 deg C) and did authors observe any change if the temperature was not controlled?
- Did CO flow above the electrolyte was maintained during electrochemical measurements?
- What is the purpose of conditioning the electrode surface in PB?
- What is the purity of CO gas used for saturation?

We thank the reviewer for these helpful suggestions. We would like to point out that the equilibration of the SPGE with the measurement solution (PB) was simply carried out to ensure that the SPGE surface was cleaned and treated with PB prior to the enzyme immobilization process. In addition, we note that a CO flow was not maintained during the electrochemical

measurements as a fresh saturated CO solution was prepared prior to these measurements. To reflect reviewer's comment, we have revised the experimental section of the manuscript to be more informative as recommended.

Page 23, lines 496–499:

“Prior to carrying out the electrochemical measurements, the electrolyte was saturated with CO (99.95%, Daedeok Gas, Korea) for 30 min, after which time, no significant pH change was observed. All electrochemical tests were conducted in a temperature-controlled room at 30 °C, unless otherwise stated.”

7. In ESI-Fig. 5b – the unit of the square root of scanning speed is missing and at which potential the peak current was taken to prepare Fig ESI-5b?

We thank the reviewer for drawing our attention to this omission. We have now added the unit of the square root of the scan rate to ESI Fig. 4b (previously Fig. 5b). In addition, we note that the peak current was recorded at 0 V and is now included in the caption. We also revised the figure numbering accordingly.

Supplementary Figure 4b

b) Linear dependence of peak current (at 0 V) with the square root of the scan rate.

8. What was the final concentration of MB in PB solution since 50 micromol MB was added. Such description indicates that the initial concentration of MB equals 50 micromol but the final one can significantly differ depending of the volume of the electrolyte.

We thank the reviewer for drawing our attention to this ambiguous description. The final concentration of MB was 50 μM. The corresponding sentence has been revised accordingly:

Page 17, lines 341:

“Thus, the enzyme–electrodes that had been previously tested for DET were tested for MET after the addition of MB (final concentration = 50 μ M) to the CO-saturated PB solution.”

9. What is the repeatability of the electrochemical activity for gold electrode modified with an enzyme?

We thank the reviewer for this comment. All electrochemical experiments were repeated three times and the data shown are mean \pm standard deviation (n=3) for all values described in relation to the electrochemical measurements.

10. In one part of the “Methods” authors write about PB with pH=8 and in the other that pH=7.2.

How pH of PBS was changed or it is just mistake in some of those values?

We appreciate the reviewer’s comment. We note that we used buffer solutions of different pH values and composition during the protein purification steps. More specifically, in the enzyme preparation step, we used 50 mM sodium phosphate buffer (pH 8), 300 mM NaCl, 10 mM imidazole, 10% glycerol and an EDTA-free protease inhibitor as our resuspension buffer. For the washing buffer, we used 50 mM sodium phosphate at pH 7.5, 300 mM NaCl, 20 mM imidazole, and 10% glycerol. Finally, we used 100 mM potassium phosphate buffer (pH 7.2) containing 300 mM NaCl, 20 mM imidazole, and 10% glycerol to elute the enzyme. We attempted the use of pH 7.2 buffer for all steps during enzyme preparation, however, the enzyme did not bind well to the resin at pH 7.2 but instead exhibited a good binding affinity at pH 8. This purification protocol was optimized for the production of CODH.

11. Authors described the significance of the obtained results as important for the fabrication of DET-based bioelectrocatalysis system. Can authors develop more the importance of their results?

We appreciate the reviewer’s comment. Our result revealed that the genetic modification of an enzyme to fuse gold-binding peptide generates a “wire-ready” enzyme having a specific affinity towards gold for soft immobilization. We proposed a straightforward approach for controlling the enzyme orientation on a solid support wherein the cofactor was within the DET distance, and this was achieved through site-specific immobilization via gold-binding peptide fusion at specific sites. We have shown that for our method to successfully result in DET, the enzyme should meet specific criteria that serve as a guideline for prospective DET studies using solid-binding peptides.

Controlling the enzyme orientation is not only critical in the context of interfacial DET in a single enzyme–electrode reaction, but also in enzyme cascade reactions where multiple enzymes should ideally adopt specific orientations for facile substrate (intermediate) channeling. Moreover, controlled enzyme orientation on an electrode surface is also important in the context of MET because the enzyme molecules should be oriented in a manner that renders the redox site accessible to both the substrate and a small redox mediator for fast electron transfer. To reflect the reviewer’s comment, the above description has been included in the revised discussion.

Page 19, lines 401–406:

“Controlling the enzyme orientation is not only critical in the context of interfacial DET in a single enzyme–electrode reaction, but also in enzyme cascade reactions where multiple enzymes should ideally adopt specific orientations for facile substrate (intermediate) channeling. Moreover, controlled enzyme orientation on an electrode surface is also important in the context of MET because the enzyme molecules should be oriented in a manner that renders the redox site accessible to both the substrate and a small redox mediator for fast electron transfer.”

12. Can author clearly indicate why electrochemical tests were not carried out for the modified bare gold disc electrode, since in my opinion the justification is unclear and not well explained? I hope that those comments will be helpful in the correction of those projects?

We thank the reviewer for these questions. In this regard, the electrochemical tests were conducted using the commercially available SPGE to eliminate any disparity when using the lab-based three-electrode measurement systems and hence ensure a good reproducibility. Although the surface morphologies of the planar gold surface and the SPGE are different, the electrochemical result revealed that only the CODH-L possessing two tethering sites (i.e., CODH-L_{gbpNC}) was available for efficient DET, which is consistent with our distance predictions and AFM results obtained using a planar gold surface. It is therefore suggested that CODH-L_{gbpNC} acquired a stable orientation, and that the orientation of CODH-L possessing gbp at either the N- or the C- terminus would be flexible due to the highly specificity of gbp toward the gold surface, either in an atomically flat or structured topography. Therefore, our estimation regarding the enzyme orientation and the ET distance do not apply to only atomically flat surfaces, but also to the SPGE surface, which is important as this also indicates that our approach is readily adaptable for use in actual SPGE-based enzyme–electrode applications. To reflect these points, the discussion has been expanded as follows:

Page 19, lines 378–386:

“It is worth noting that although the surface morphologies of the planar gold surface and the SPGE are different, the electrochemical result revealed that only the CODH-L possessing two tethering sites (i.e., CODH-L_{gbpNC}) was available for efficient DET, which is consistent with our distance predictions and AFM results obtained using a planar gold surface. It is therefore suggested that CODH-L_{gbpNC} acquired a stable orientation, and that the orientation of CODH-L possessing gbp at either the N- or the C- terminus would be flexible due to the highly specificity of gbp toward the gold surface, either in an atomically flat or structured topography. Therefore, our estimation regarding the enzyme orientation and the ET distance do not apply to only atomically flat surfaces, but also to the SPGE surface.”

REVIEWERS' COMMENTS:

Reviewer #1 (Remarks to the Author):

I have no further comments

Reviewer #2 (Remarks to the Author):

I am satisfied with the responses and changes made by the authors.